On the diversity of the SE Indo-Pacific species of Terebellides (Annelida; Trichobranchidae), with the description of a new species

Parapar Julio 1
Moreira Juan 2
Martin Daniel dani@ceab.csic.es 3
1 Departamento de Bioloxía Animal, Bioloxía Vexetal e Ecoloxía, Universidade da Coruña , A Coruña , Spain
2 Departamento de Biología (Zoología), Facultad de Ciencias, Universidad Autónoma de Madrid , Cantoblanco , Spain
3 Department of Marine Ecology, Center for Advanced Studies of Blanes (CEAB—CSIC) , Blanes , Catalunya , Spain
Osborn Karen
Electronic publication date: 2016 Aug 16
Publication date: 2016
Volume: 4
Electronic Location ID: e2313
Received 2016 May 9; Accepted 2016 Jul 11
Copyright: ©2016 Parapar et al.
Copyright year: 2016
Copyright holder: Parapar et al.
License: This is an open access article distributed under the terms of the Creative Commons Attribution License, which permits unrestricted use, distribution, reproduction and adaptation in any medium and for any purpose provided that it is properly attributed. For attribution, the original author(s), title, publication source (PeerJ) and either DOI or URL of the article must be cited.
License URL: https://creativecommons.org/licenses/by/4.0/

Keywords: Polychaeta, Terebellides, New species, Branchial morphology, SEM, Thailand, Myanmar

Funding: Fauna Ibérica: Polychaeta VI: Palpata, Canalipalpata I CGL2014-53332-C5-3-P MarSymBiomics CTM2013-43287-P Ministerio de Economía y Competitividad of Spain GRC of Marine Benthic Ecology 2014SGR120 Generalitat de Catalunya The study has been financed by the research projects “Fauna Ibérica: Polychaeta VI: Palpata, Canalipalpata I” (CGL2014-53332-C5-3-P) and “MarSymBiomics” (CTM2013-43287-P), funded by the Ministerio de Economía y Competitividad of Spain, as well as by the Consolidated Research Group on Marine Benthic Ecology (2014SGR120), funded by the Generalitat de Catalunya, and the research contract between the CEAB (CSIC) and the French company Créocéan. The members of Créocean contributed to the sampling design and sample collection and explicitly agreed to allow the publication of the information contained in this study. The funders had no role in study design, data collection and analysis, decision to publish, or preparation of the manuscript.

==============================
The study of material collected during routine monitoring surveys dealing with oil extraction and aquaculture in waters off Myanmar (North Andaman Sea) and in the Gulf of Thailand, respectively, allowed us to analyse the taxonomy and diversity of the polychaete genus Terebellides (Annelida). Three species were found, namely Terebellides cf. woolawa, Terebellides hutchingsae spec. nov. (a new species fully described and illustrated), and Terebellides sp. (likely a new species, but with only one available specimen). The new species is characterised by the combination of some branchial (number, fusion and relative length of lobes and papillation of lamellae), and thoracic (lateral lobes and relative length of notopodia) characters and is compared with all species described or reported in the SW Indo-Pacific area, as well as with those sharing similar morphological characteristics all around the world. The taxonomic relevance of the relative length of branchial lobes and different types of ciliature in branchial lamellae for species discrimination in the genus is discussed. A key to all Terebellides species described in SE Indo-Pacific waters is presented.

Introduction

The genus Terebellides is characterised by a combination of several characters including the compact appearance of the prostomium, a peristomium forming two lips (upper and lower), a thorax composed by 18 chaetigers, capillary notochaetae, denticulate thoracic neurochaetal hooks and abdominal avicular uncini. Nevertheless, the two most distinctive characters are the single mid-dorsal branchiae composed by 2–5 lamellate lobes, and the geniculate chaetae present in the first 1–2 thoracic neuropodia (Schüller & Hutchings, 2013).

The peculiar shape of the branchiae of the type species (i.e., T. stroemii Sars, 1835) was one of the main facts that led to attributing most of the subsequent records to this taxon, especially in the 19th century and most part of the 20th, when only a few characters were enough to discriminate the different species. Therefore, the number of fully described species was relatively low and T. stroemii was thought as being cosmopolitan. The ‘Catalogue of World Polychaetes’ by Hartman (1959) contributed to this consideration by synonymizing several species with T. stroemii (e.g., T. ypsilon). As a result of this, up to 1980s this species was reported from a wide variety of world areas and depths. However, in the late 20th century, this set of characters was gradually revealed not to be enough to distinguish among species. The study done by Williams (1984) clearly showed that a new set of characters was needed to differentiate species. Imajima & Williams (1985) and Solis-Weiss, Fauchald & Blankestein (1991) further supported this trend and, thus, a progressively higher number of new species have been described (e.g., Bremec & Elías, 1999; Hilbig, 2000; Hutchings & Peart, 2000; Garraffoni & Lana, 2003; Hutchings, Nogueira & Carrerette, 2015; Parapar & Moreira, 2008; Parapar, Moreira & Helgason, 2011; Parapar, Mikac & Fiege, 2013; Parapar, Moreira & O’Reilly, 2016; Parapar et al., 2016; Schüller & Hutchings, 2010; Schüller & Hutchings, 2012; Schüller & Hutchings, 2013). At the same time new characters for the species discrimination have been reported, and those traditionally used (e.g., branchial shape) have increasingly been described in greater detail. As a result, the true diversity of the genus Terebellides begins to be revealed.

In the SW Indo-Pacific, ten species of Terebellides have been described: four from the Philippine and China Seas (Salazar-Vallejo et al., 2014), namely T. intoshi Caullery, 1915, T. jorgeni Hutchings, 2007, T. sieboldi Kinberg, 1866 and T. ypsilon Grube, 1878, and six from the Australian coasts: T. akares Hutchings, Nogueira & Carrerette, 2015, T. jitu Schüller & Hutchings, 2010, T. kowinka Hutchings & Peart, 2000, T. mundora Hutchings & Peart, 2000, T. narribri Hutchings & Peart, 2000 and T. woolawa Hutchings & Peart, 2000. Additional references to the presence of T. stroemii in these waters are found in Caullery (1944), Rullier (1965), Gallardo (1967), Stephenson, Williams & Lance (1970), Stephenson, Williams & Cook (1974), Gibbs (1971), Knox & Cameron (1971), Hutchings (1977), Shin (1982), Amoureux (1984), Hutchings & Murray (1984), Hutchings et al. (1993) and Tan & Chou (1993). In Australia and New Zealand, reports of T. stroemii were summarized by Day & Hutchings (1979). Later, Hutchings & Peart (2000) reviewed the Australian Terebellides and described four new species. Moreover, these authors also reviewed materials collected in the vicinity of the type locality in the SW coast of Norway and concluded that T. stroemii is not present in southern latitudes. Further papers by Hutchings (2007), Schüller & Hutchings (2010) and Hutchings, Nogueira & Carrerette (2015) continued with the reassessment of the diversity of Terebellides in Australian-Indonesian coasts.

Our paper addresses the study of the genus in waters off Myanmar and Thailand, allowing us to describe a new species. We are also reviewing and updating the previous works reporting this genus in the area, and we present a key to all species recorded in the SE Indo-Pacific. Our study represents one additional contribution to unveil the hidden diversity of the genus Terebellides in the world oceans and confirms that T. stroemii appears to be absent in the Indo-Pacific. Furthermore, we provide evidence supporting that the diversity of Terebellides is still far from being well known. This seems to be particularly true for the Indo-Pacific region as, in addition to the new one, we are also describing two more species. Despite that its particular situation (either in terms of morphology or number of available specimens) does not allow us to fully identify them, they are certainly different, thus contributing to the overall diversity of the genus in the region.

Material and Methods

This study is based on 85 specimens of the genus Terebellides from 26 samples collected in 21 stations during routine monitoring surveys dealing with oil extraction and aquaculture in waters off Myanmar (North Andaman Sea) in 2003 and Gulf of Thailand in 1998, respectively (Table 1).

Table 1 Main abiotic characteristics: silt and organic carbon contents.

Main abiotic characteristics of the samples where Terebellides specimens were collected. Org. car., organic carbon content (%).

	Station	Date	Longitude N	Latitude E	Depth (m)	≤ 63 µm%	Org. car.	
Myanmar (N Andaman Sea)	E7	03/12/03	15°07′59.8″	94°46′46.5″	46.0	76.3	1.26	
E8	“	15°06′27.7″	94°46′50.2″	47.0	78.7	1.29	
E11	“	15°07′23.2″	94°46′45.9″	47.0	83.6	1.21	
E14	“	15°07′20.0″	94°46′51.1″	47.4	76.7	1.12	
E15	“	15°07′15.9″	94°46′51.1	47.0	80.4	1.16	
E16	“	15°07′07.8″	94°46′51.0″	47.6	73.8	1.15	
E17	“	15°07′14.9″	94°45′25.9″	48.0	74.0	1.28	
S2	“	15°02′03.3″	94°45′45.7″	51.0	72.8	0.86	
S3	“	15°02′19.4″	94°46′02.6″	51.0	80.5	0.98	
S4	“	15°03′08.2″	94°46′03.0″	51.0	90.2	0.92	
WP2	“	15°09′06.6″	94°45′26.7″	45.5	86.2	1.19	
WP3	“	15°02′03.0″	94°46′19.2″	51.0	69.6	2.42	
Gulf of Thailand	2	17/07/98	07°27′43.6″	102°39′00.1″	61.0	88.3	0.81	
3	“	07°29′20.2″	102°43′49.7″	61.0	73.6	0.74	
5	“	07°36′01.1″	102°50′32.6″	64.0	92.2	1.15	
6	“	07°40′08.9″	102°51′14.6″	66.0	95.4	1.23	
7	“	07°34′00.8″	102°45′37.2″	64.0	89.3	0.98	
8	“	07°35′23.6″	102°44′12.3″	66.0	92.7	1.20	
15	“	07°44′21.5″	102°34′58.5″	75.0	93.8	1.42	
16	“	07°43′18.8″	102°28′39.1″	77.0	90.7	1.22	
23	“	08°01′12.7″	102°18′40.2″	78.0	75.6	1.08	

The samples were collected by means of a van Veen grab covering about 0.3 m2. The grab contents were mixed in a sufficiently large container, and then sieved out on board by pouring the contents through a 1 mm mesh sieve. The retained sediment was then transferred into a plastic bag, fixed with a 10% formaldehyde/seawater solution, stained with “Rose of Bengal” and stored until sorted. An initial sorting was performed under a dissecting stereomicroscope (Zeiss Stemi 2000-C) and the specimens of Terebellides were counted and preserved in 70% ethanol.

One-liter volume of sediment from one grab was used for physico-chemical analyses (viz. granulometry, organic carbon content). The sediment was taken at each station and transferred to wide-mouthed double-closing 500 ml polyethylene flasks, which were stored in the dark until transferred to the laboratory. Laser granulometry (% volume) was performed on dry sediment after sifting through a 0.8 mm mesh sieve using a Malvern Mastersizer S laser granulometer. Sediments were characterized by the percentage of silt and clay (diameter <63 µm). Estimates of organic carbon have been made according to the European experimental standard NF ISO 14235 (oxidation method, 0.1% m/m).

Light microscope images were obtained by means of an Olympus SZX12 stereomicroscope equipped with an Olympus C-5050 digital camera. Line drawings were made by means of an Olympus BX40 stereomicroscope equipped with camera lucida. Specimens used for examination with Scanning Electron Microscope (SEM) were prepared by critical point drying, covered with gold and examined and photographed under a JEOL JSM-6400 electron microscope at the Servizos de Apoio á Investigación-SAI (Universidade da Coruña-UDC, Spain).

Most of the obtained material was deposited in the Museo Nacional de Ciencias Naturales (Madrid, Spain; MNCN). Additional paratypes of T. hutchingsae spec. nov. were deposited in the collections of the Australian Museum (Sydney, Australia; AM) and Göteborgs Naturhistoriska Museum (Göteborg, Sweden; GNM). Type material of Terebellides gracilis Malm, 1874 was loaned for study from the Göteborgs Naturhistoriska Museum (Holotype, GNM Polych 641). Type material of Terebellides sieboldi Kinberg, 1866 was requested to the Swedish Museum of Natural History for comparison but only one specimen, and badly preserved, could be located (L Gustavsson, in litt.).

Methyl green staining pattern was determined based on the classification proposed by Schüller & Hutchings (2010).

The electronic version of this article in Portable Document Format (PDF) will represent a published work according to the International Commission on Zoological Nomenclature (ICZN), and hence the new names contained in the electronic version are effectively published under that Code from the electronic edition alone. This published work and the nomenclatural acts it contains have been registered in ZooBank, the online registration system for the ICZN. The ZooBank LSIDs (Life Science Identifiers) can be resolved and the associated information viewed through any standard web browser by appending the LSID to the prefix http://zoobank.org/. The LSID for this publication is: urn:lsid:zoobank.org:pub:39745D2F-9163-48B2-9FAB-FBF66D3AEFB5. The online version of this work is archived and available from the following digital repositories: PeerJ, PubMed Central and CLOCKSS.

Results

Systematic account

Family Trichobranchidae Malmgren, 1866	
Genus Terebellides Sars, 1835, emended by Schüller & Hutchings, 2013	
Type species: Terebellides stroemii Sars, 1835	

Terebellides hutchingsae spec. nov.

LSID: 78E96984-41E7-43E6-8E5D-03E9421BE306	
(Figs. 1– 8 and Table 2)	

Figure 1 Line drawings of the species of Terebellides.

Terebellides hutchingsae spec. nov., Holotype, MNCN 16.01/17428. (A) anterior end, left lateral view. Terebellides sp., MNCN 16.01/17457; (B) detail of position of geniculate chaetae in thoracic chaetiger 5 (TC5) and thoracic chaetiger 6 (TC6) of anterior right side of body.

Table 2 Comparison of Terebellides with ciliated papillae in branchial lamellae.

Comparison of several body characters of the species of Terebellides described with ciliated papillae in branchial lamellae.

	Source	GP	LL	Branchiae	CH1	CHG	NRTU	NACH	Distribution	
				Lobes	Relative length	LCP/WDCP	PPP						
T. gracilis Malm, 1874	Parapar, Moreira & Helgason (2011)a	TC4–5	TC1–5b	4c	Same	LCP	NO	Shorter	6	2d	44	Iceland	
	Parapar, Mikac & Fiege (2013)	TC4–5	TC1–6	4	Ventral shorter	LCP	Yes, in lower lobes	Shorter	6	2–4	30–32	Adriatic Sea	
T. jorgeni Hutchings, 2007	Hutchings (2007)	TC 2–3e	SG1–5 and SG7–8	4	Ventral shorter	WDCPf	Nog	Shorterh	6	4–6	∼48	Balii	
T. mediterranea Parapar, Mikac & Fiege, 2013	Parapar, Mikac & Fiege (2013)	TC4–5	TC1–5	5	Ventral much shorter	LCP	Yes	Longer	6	3–4	32	Adriatic Sea	
T. akares Hutchings, Nogueira & Carrerette, 2015	Hutchings, Nogueira & Carrerette (2015)	TC4–5	TC1–8	5	Ventral shorter	WDCP	Nog	Shorter	5–6	4g	17–25	North-east Australia	
T. shetlandica Parapar, Moreira & O’Reilly, 2016	Parapar, Moreira & O’Reilly (2016)	TC4–5	TC1–6	5	Ventral shorter	WDCP	Short	Shorter	6	4	27–36	Shetland Islands	
T. persiae Parapar et al., 2016	Parapar et al. (2016)	TC4–5	TC1–6	5	Ventral shorter	WDCP	Short	Shorter	6	2–4	27–36	Iran	
T. hutchingsae spec. nov.	This work	TC4–5	TC1–5	5	Ventral shorter	WDCP	Short	Shorter	6	4	27–30	Thailand and Myanmar	
Notes.

a From a redescription of Malm’s holotype.

b Very low lappets.

c The fifth lobe is very low and therefore branchiae could be described as with only four lobes.

d Character observed from the study of several specimens under the SEM; it cannot be properly observed in holotype.

e Author mentions SG3–5; this can be interpreted as NO in SG3 (TC1) and GO in SG4–5 (TC2–3); author probably meant TC3–5.

f Author simply mentions “surface of (branchial) lamellae weakly papillate”.

g Not explicitly mentioned in the original description of the species; taken from figures.

h CH2 also smaller than subsequent ones.

i This species shows a disjunct distribution, being originally described from Bali but also reported from W and S Africa and Tasman Sea.

Figure 2 Stereomicroscope images of Terebellides spp.

(A–B) Terebellides hutchingsae spec. nov. Stereomicroscope images. Holotype MNCN 16.01/17428; (A) general view; (B) anterior end, left lateral view. (C–D) Terebellides cf. woolawa MNCN 16.01/17455; (C) anterior end right lateral view; (D) detail of branchiae, left lateral view. (E–F) Terebellides sp. MNCN 16.01/17457; (E) anterior end right ventro-lateral view, arrowheads pointing to thoracic chaetiger 5 and thoracic chaetiger 6; (F) detail of branchiae in ventral view. Abbreviations: BL, branchial lobes; BT, buccal tentacles; TC, thoracic chaetiger; tp, terminal projection.

Material examined

THAILAND (Gulf of Thailand): Holotype: MNCN 16.01/17428 (St. 6). Paratypes: MNCN 16.01/17429 (St. 2, 4 specs); MNCN 16.01/17430 (St. 3, 3 specs); MNCN 16.01/17431 (St. 5, 3 specs); MNCN 16.01/17432 (St. 5, 1 spec. on SEM stub); MNCN 16.01/17433 (St. 6, 5 specs); MNCN 16.01/17434 (St. 7, 5 specs); MNCN 16.01/17435 (St. 8, 7 specs); MNCN 16.01/17436 (St. 8, 1 spec. on SEM stub); MNCN 16.01/17437 (St. 15, 2 specs); MNCN 16.01/17438 (St. 16, 6 specs); MNCN 16.01/17439 (St. 23, 1 spec. on SEM stub). MYANMAR (North Andaman Sea): Paratypes: MNCN 16.01/17440 (St. E7(2), 1 spec.); MNCN 16.01/17441 (St. E8(3), 1 spec.); MNCN 16.01/17442 (St. E11B(2), 4 specs); MNCN 16.01/17443 (St. E11B(3), 2 specs); MNCN 16.01/17444 (St. E14(2), 4 specs); MNCN 16.01/17445 (St. E15(2), 10 specs); MNCN 16.01/17446 (St. E16(1), 2 specs); MNCN 16.01/17447 (St. E16(3), 1 spec.); MNCN 16.01/17448 (St. 17(3), 1 spec.); MNCN 16.01/17449 (St. S2(2), 1 spec.); MNCN 16.01/17450 (St. S3(2), 4 specs); MNCN 16.01/17451 (St. S3(2), 1 spec. on SEM stub); MNCN 16.01/17452 (St. S3(3), 4 specs); MNCN 16.01/17453 (St. S4(2), 2 specs); GNM Polychaeta 14880 (St. S4(3), 1 spec.); GNM Polychaeta 14881 (St. WP2(2), 2 specs); AM 48834 (St. WP2(3), 2 specs); MNCN 16.01/17454 (St. WP2(3), 2 specs on SEM stub); AM 48835 (St. WP3(3), 1 spec.).

Figure 3 SEM micrographs of paratypes of Terebellides hutchingsae spec. nov. from Thailand.

Terebellides hutchingsae spec. nov., SEM micrographs of paratypes, MNCN 16.01/17439 and MNCN 16.01/17436. (A) anterior end, left lateral view, showing lateral lappets and relative size of branchial lobes (BL); (B) detail of branchial lobes showing degree of fusion between dorsal (DL) and ventral (VL) lobes, framed area showed in (C); (C) detail of distal end of left dorsal lobe showing ciliary fields and terminal projection (TP); (D) right lateral view of branchiae, framed area showed in (E); (E–F) detail of ciliated papillae of branchial lobes lamellae.

Figure 4 SEM micrographs of paratypes from Thailand and Myanmar.

Terebellides hutchingsae spec. nov., SEM micrographs of paratypes, MNCN 16.01/17432 and MNCN 16.01/17454. (A) several buccal tentacles, one showing ciliated side of distal end (arrowhead); (B) detail of ciliated distal end of a buccal tentacle; (C) anterior end, left lateral view, showing lateral lappets (LL) from thoracic chaetiger 1 (TC1) to thoracic chaetiger 6 (TC6) and nephridial papilla (NPA) in thoracic chaetiger 1; (D) detail of genital opening (GO) of thoracic chaetiger 4 (TC4) and thoracic chaetiger 5 (TC5); (E) thoracic chaetiger 1 (TC1) and thoracic chaetiger 2 (TC2) notopodia showing reduction of TC1; (F) detail of pilose surface of thoracic notochaetae.

Description (based on holotype and paratypes) Complete individuals ranging from 9.0 to 14.0 mm in length (14 mm in holotype; Figs. 1A and 2A–2B) and 0.7 to 1.5 mm in maximum width at thoracic region (1.3 mm in holotype, excluding parapodia). Body tapering posteriorly with segments increasingly shorter and crowded towards pygidium. Prostomium compact; peristomium forming a tentacular membrane with large upper and lower lips surrounding mouth, sometimes almost devoid of buccal tentacles (Fig. 3A). Buccal tentacles of two types, short ventral tentacles uniformly cylindrical or slightly expanded at tips, and long dorsal tentacles more expanded at tips (Figs. 2B and 4A–4B). Lateral lappets on TC1–5 (SGIII–VII), being larger in TC1–3 (Figs. 2B, 3A, 4C and 6A ). No conspicuous dorsal rounded projection on anterior chaetigers or oval-shaped glandular region in TC3. Both notopodia and notochaetae in TC1 less developed than in following chaetigers (Figs. 3A, 4C and 4E).

Branchiae arising as single structure from SGII–III, with a single, mid-dorsal, stalk and two pairs of unfused lobes; lower (=ventral) (BL3–4) pair smaller and much shorter than upper (=dorsal) (BL1–2) pair of lobes (Figs. 3A–3B and 6B–6C). Upper and lower lobes with a short terminal pointed projection (although deciduous and sometimes damaged) (Fig 3C). Dorsal pair of branchial lobes with short anterior projection (fifth lobe; BL5) (Fig. 3D), sometimes hidden behind buccal tentacles (Figs. 2A and 2B). Loss of any of branchial lobes not observed. One side of branchial lamellae with parallel bent rows of cilia and well- developed ciliated papillae on edge of one side of each branchial lamella (Figs. 3D–3F).

Figure 5 SEM micrographs of paratypes from Thailand and Myanmar, Cont.

Terebellides hutchingsae spec. nov., SEM micrographs of paratypes, MNCN 16.01/17432 and MNCN 16.01/17454. (A) row of uncini in a thoracic uncinigerous torus in latero-frontal view; (B) detail of two thoracic uncini in lateral view; (C) detail of a thoracic uncinus in frontal view showing rostrum (R) and four teeth forming first row of capitium denticles; (D) one abdominal neuropodial pinnule; (E) one abdominal uncinus in upper view showing first row of capitium denticles almost reaching tip of rostrum (R); (F) four abdominal uncini in upper view.

Figure 6 SEM micrographs of paratypes from Myanmar.

Terebellides hutchingsae spec. nov., SEM micrographs of paratypes, MNCN 16.01/17451 and MNCN 16.01/17454. (A) anterior end, left lateral view, showing lateral lappets (LL) from thoracic chaetiger 1 (TC1) to thoracic chaetiger 6 (TC6) and geniculate chaetae (GC) in thoracic chaetiger 6; (B) lateral view of lower branchial lobes (BL3–4) showing terminal projection (TP); (C) ventral view of lower branchial lobes (BL3–4) with terminal projection (TP); (D) dorsal view of thoracic chaetiger 4 (TC4) to thoracic chaetiger 7 (TC7) showing genital openings (GO) at TC4 and TC5, and geniculate chaetae (GC) at TC6; (E) thoracic chaetiger 6 geniculate chaetae in lateral view; (F) detail of capitium of one geniculate chaeta.

Eighteen thoracic chaetigers (SGIII–XX), all with notopodia; neuropodia from SGVIII. Notopodia of TC1 smaller than following ones (Figs. 4C and 4E); all remaining notopodia similar in size. Thoracic neuropodia as sessile pinnules, from TC6 (SGVIII) to TC18 (SGXX), with uncini in single rows from TC7 (SGIX) throughout. Thoracic notochaetae similar in length, with textured surface (Fig. 4F). Ciliated papilla dorsal to each thoracic notopodia not observed. First thoracic neuropodia (TC6) with 4–7 geniculate acicular chaetae with minute teeth in their upper part forming a capitium easily overlooked without SEM (Figs. 6E–6F); sharply bend. Subsequent thoracic neuropodia with one row of about 8–10 uncini per torus (Fig. 5A); uncini as shafted denticulate hooks with long, pointed rostrum surmounted by 4–5 teeth and an upper crest of several smaller denticles of different sizes (Figs. 5A–5C). One finger-shaped nephridial papilla basal to branchial stem (Fig. 4E); genital openings, dorsal to notopodia in TC4 and TC5 ( Figs. 4D and 6D).

Twenty seven to 30 abdominal chaetigers (30 in holotype). Abdominal neuropodia as erect pinnules, with about 30 uncini per torus (Fig. 5D). Uncini with 3–4 teeth above rostrum (Figs. 5E and 5F), surmounted by a row of an irregular number of shorter teeth and an upper crest of minute teeth. Pygidium blunt, funnel-like depression. No eggs were observed in body cavity of holotype, but mature females of smaller size were observed (9.0 mm length, 1.0 mm width). Colour in alcohol pale brown.

Methyl green staining ressembling pattern 1 of Schüller & Hutchings (2010), resulting in a solid green coloration from CH1 to CH9, then turning into striped pattern from CH10 to CH12 and fading in following segments. Additional pronounced staining also on short ventral branchial lobes (BL3–4) and postero-ventral part of branchial upper lobes (BL1–2).

Type locality

Gulf of Thailand, muddy bottom at 66 m depth.

Distribution and habitat

Specimens of T. hutchingsae spec. nov. were found in shallow water bottoms (45.5–51.0 m depth) about 80 km off the coast of Myanmar (North Andaman Sea) and in slight deeper bottoms (61.0–78.0 m depth) in the outer end of the Gulf of Thailand (Table 1 and Fig. 7). Sediments are typically muddy with high values of organic carbon (0.74–2.42%) and of silt and clay fraction (69.6%–95.4%).

Etymology

The species is named after Dr. Pat Hutchings, for her many contributions to the taxonomy of Terebelliform polychaetes in Australia and SW Pacific waters, and particularly to the genus Terebellides, and also for her key role in the study of Australian polychaetes.

Terebellides cf. woolawa Hutchings & Peart, 2000

(Figs. 2C–2D and 7)	

Material examined

MNCN 16.01/17455 (St. S4(3), 1 spec.); MNCN 16.01/17456 (St. WP3(3), 1 spec.).

Distribution and habitat

Both specimens of T. cf. woolawa were found in two shallow water stations (51.0 m depth) about 80 km off the mouth of the Irawadi river in the coast of Myanmar (North Andaman Sea) (Table 1).

Terebellides sp.

(Figs. 1B, 2E–2F and 7)	

Material examined

MNCN 16.01/17457(St. S4(3), 1 spec.).

Distribution and habitat

The specimen was found in a shallow water bottom (51 m depth) about 16 km off the coast of Myanmar (North Andaman Sea) (Table 1).

Figure 7 Map of SW Indo-Pacific Ocean.

Map of SW Indo-Pacific Ocean showing locations where species of Terebellides reported in this paper were found (big circles), along with type localities of other Terebellides species previously described (medium circles) or reported (small circles).

Key of SE Indo-Pacific species of Terebellides

The key here presented has been modified from a previous key of Australian Trichobranchidae by Hutchings & Peart (2000), which was based on a limited number of easy-to-detect characters: (1) number of chaetigers with geniculate chaetae, (2) degree of development of thoracic notopodia, and (3) shape of branchiae, giving special emphasis to the relative size of branchial lobes. Terebellides ypsilon Grube, 1878, from the Philippines, was not included because the description is very brief and following Hutchings & Peart (2000), who revised the type material, the taxon should be considered as undeterminable.

1. Geniculate chaetae in two thoracic chaetigers	2	
- Geniculate chaetae in one thoracic chaetiger	4	
2. All thoracic notopodia of similar length	3	
- Thoracic notopodia from thoracic chaetiger 6 onwards much bigger and with more numerous and longer notochaetae	T. intoshi Caullery, 1944	
3. Thoracic uncinigers with geniculate chaetae similar in shape and position1	T. akares Hutchings, Nogueira & Carrerette, 2015	
- Thoracic uncinigers with geniculate chaetae different in shape and position	Terebellides sp.	
4. Geniculate chaetae in thoracic chaetiger 72	T. sieboldi Kinberg, 1866	
- Geniculate chaetae in thoracic chaetiger 6	5	
5. Branchial lobes 1–4 not fused	T. mundora Hutchings & Peart, 2000	
- Branchial lobes 1–4 more or less fused	6	
6. Four branchial lobes	7	
- Five branchial lobes	8	
7. All thoracic notopodia similar in size and well developed	T. kowinka Hutchings & Peart, 2000	
- Thoracic notopodia1 and 2 much smaller than subsequent ones	T. jorgeni Hutchings, 2007	
8. Branchial lobe 5 about 1/5 length of posterior lobes; thoracic lateral lappets without dorsal projections, geniculate chaetae of thoracic chaetiger 6 sharply bent	9	
- Branchial lobe 5 almost 1/2 length of posterior lobes; lateral lappets of thoracic chaetigers 1–4 with dorsal projections, geniculate chaetae of thoracic chaetiger 6 gently curved	T. woolawa Hutchings & Peart, 2000	
9. Thoracic notopodia1 not reduced; large, white, oval glandular patches in thoracic notopodia 3	T. narribri Hutchings & Peart, 2000	
- Thoracic notopodia1 strongly reduced; no glandular patches in thoracic notopodia 3	10	
10. All branchial lobes of similar length and fused half of their length; branchial lamellae with transverse ridges of ciliature	T. jitu Schüller & Hutchings, 2010	
- Ventral (lower) branchial lobes much shorter than dorsal (upper) ones and fused basally; branchial lamellae with ciliated transverse ridges of ciliature and ciliated papillae	T. hutchingsae spec. nov.	

(1) Those of first pair (segment 7) are shorter.

(2) The position of GC in TC7 (SG9) is very unusual in the genus Terebellides; this feature is apparently only shared with T. pacifica Kinberg, 1866, a species which has been removed from synonymy with T. stroemii by Garraffoni, Lana & Hutchings (2005).

Discussion

On Terebellides hutchingsae spec. nov.

Five species of Terebellides were previously described in the equatorial Indo-Pacific region (Fig. 7): T. intoshi Caullery, 1915, T. sieboldi Kinberg, 1866, T. ypsilon Grube, 1878, T. jorgeni Hutchings, 2007 and T. jitu Schüller & Hutchings, 2010. Terebellides intoshi is characterised by the large size of the notopodia and notochaetae from TC6 onwards (Fig. 8A ) and probably by the presence of two chaetigers with geniculate chaetae as well (see also the discussion on Terebellides sp.); T. sieboldi has geniculate chaetae in TC7 (SG9) instead of TC6 and T. ypsilon is considered undeterminable by Hutchings & Peart (2000) because type material no longer exists. The two most recently described species, Terebellides jorgeni and T. jitu, are the most similar to T. hutchingsae spec. nov. The overall shape of branchiae is quite similar in T. jorgeni and T. hutchingsae sp. nov., being lobes 1–4 unequally sized and entirely free (not fused), with upper (dorsal) ones larger than lower (ventral) ones, and with “surface of branchial lamellae weakly papillate” (cfr. p. 78 in Hutchings, 2007); the latter probably refers to the presence of ciliated papillae, which is a feature difficult to confirm in the original figures. On the other hand, T. jorgeni differs from the new species in: (1) the presence of glandular and whitish ventral part of anterior segments, SG5 to SG9 (CH3 to CH7) but specially on SG5 to SG7 (absent in T. hutchingsae sp. nov.), and bearing pronounced thickening and elevation of dorsal anterior margins forming dorsal crests; (2) genital openings are present in SG4 and SG5, instead of SG6 and SG7 (TC4 and TC5) as in T. hutchingsae spec. nov.; (3) the branchiae are formed by four lobes instead of five.

According to the original description, T. jitu is also similar to T. hutchingsae spec. nov. but all branchial lobes are of similar length and have half of their length fused, instead of the lower ones being much shorter and fused basally as in T. hutchingsae spec. nov.

Terebellides narribri Hutchings & Peart, 2000 and T. woolawa Hutchings & Peart, 2000 were described from the NE Australian coast. Both species share with T. hutchingsae spec. nov. branchiae with similar shape and composed by five lobes; T. narribri differs from the new species by having first thoracic notopodia (TN1) of same size as the following, and TC3 bearing large, white, oval pair of glandular patches. Terebellides woolawa is characterised by the great development of BL5 (but see discussion on T. cf. woolawa) and by having anterior thoracic segments with dorsal projections on lateral lappets, which are absent in T. hutchingsae spec. nov.

Out of the Indo-Pacific area, there are no reports yet of any species gathering the same set of characters as T. hutchingsae spec. nov. However, seven more species have been described with branchial lamellae having ciliated papillae (Solís-Weiss, Fauchald & Blankestein, 1991; Bremec & Elias, 1999; Parapar et al., 2016). These are T. klemani Kinberg, 1867, T. anguicomus Müller, 1858, T. carmenensis Solís-Weiss, Fauchald & Blankestein, 1991, T. parvus Solís-Weiss, Fauchald & Blankestein, 1991, T. lanai Solís-Weiss, Fauchald & Blankestein, 1991, T. totae Bremec & Elías, 1999 and T. persiae Parapar et al., 2016. Four of them differ from our new species in having “thoracic dorsal hump”: T. anguicomus (which also has 17 thoracic chaetigers instead of the 18 in T. hutchingsae spec. nov.), T. carmenensis, T. totae (which also has all four branchial lobes similar in length, while the ventral ones are much shorter in T. hutchingsae spec. nov.) and T. persiae. As for the three remaining species, T. klemani has all branchial lobes of similar length and free almost to base, and gently curved geniculate acicular chaetae instead of being sharply bent, T. lanai has only one type of buccal tentacles, instead of two, and branchial lobes are fused in most of their length; and T. parvus is characterized by having a very low number of abdominal segments (20–26 instead of 27–30 in our new species).

Finally, two species show tufts of cilia at the edge of the branchial lamellae: T. mediterranea Parapar, Mikac & Fiege, 2013, and T. gracilis (Malm, 1874) both from Iceland, sensu Parapar, Moreira & Helgason (2011), and from the Adriatic sea, sensu Parapar, Mikac & Fiege (2013), but in these two species the cilia arise directly from the lamella surface and not from the tip of a papilla as in T. hutchingsae spec. nov.

On Terebellides cf. woolawa and Terebellides sp.

Terebellides woolawa is characterised by the well-developed fifth branchial lobe (BL5) and the presence of dorsal rounded projections on lateral lappets of SG 3–6 (TC1–4). This large species was described from intertidal to shallow water habitats in eastern Australia (Fig. 7) and was found across most of Australian coasts (Hutchings & Peart, 2000). Specimens found in this study are large-sized, and agree fairly well with the original description; in particular, specimen MNCN 16.01/17455 shows the typical shape of the branchiae, which have five lobes, BL1–4 are fused up to half of their length, filamentous tips are short, and BL5 is well developed (Figs. 2C and 2D). Nevertheless, our specimens lack the characteristic dorsal lobes of anterior thoracic lateral lappets: this prevented to fully confirm the identity of our material.

In turn, the specimen identified as Terebellides sp. differs from T. hutchingsae spec. nov. and Terebellides cf. woolawa in two features: (1) BL5 is large-sized, about half the length of posterior lobes (BL1–4); and (2) TC5 and TC6 are both provided with acicular geniculate chaetae. Thus, BL5 is longer than in any other described species including T. woolawa. However, this might be due to the preservation state of the specimen, which is slightly deteriorated. Anyway, the combination of the two aforementioned characters may justify the erection of a new species, but we prefer to wait for eventually finding additional specimens to confirm its status.

On the other hand, four species of Terebellides were previously described as having geniculate chaetae in two thoracic chaetigers. Two species are known from the French Polynesia and Iceland: T. biaciculata Hartmann-Schröder, 1992 and T. bigeniculatus Parapar, Moreira & Helgason, 2011, respectively. The other two belong to the Indo-Pacific region: T. akares Hutchings, Nogueira & Carrerette, 2015 and T. intoshi Caullery, 1945 sensu Imajima & Williams (1985), from North-East Australia and Japan, respectively. We follow Parapar, Moreira & Helgason (2011) in considering that type material of T. intoshi from South China Sea probably does not have two chaetigers with geniculate chaetae. Thus, the Japanese material would belong to a different species. Anyway, they differ from Terebellides sp. in the branchial shape and the greater development of thoracic notopodia from TC6 (Fig. 8A). In turn, T. akares, have branchiae bearing a much shorter BL5 and the posterior ventral lobes (BL3–4) are completely free from each other, contrary to Terebellides sp., in which these lobes are fused in most of their length (Fig. 2F). The observed differences between these Indo-Pacific species and Terebellides sp. reinforce is status as new species, so that further efforts should be addressed to find new materials allowing to formally describe it.

Figure 8 Line drawings of two Terebellides species previously described or reported in SW Pacific Ocean by Caullery (1915), redrawn from original.

Terebellides intoshi Caullery, 1915; (A) anterior end in right lateral view showing great size of notopodia and notochaetae of thoracic chaetigers from thoracic chaetiger 6 onwards. Terebellides stroemii Sars, 1835; (B) anterior end in left lateral view; (C–D) dorsal and ventral view of branchial lobes showing high development of fifth branchial lobe (BL5), small size of branchial ventral lobes (BL3–BL4) and high degree of fusion of all posterior lobes (BL1–BL4).

On Terebellides stroemii in Indo-Pacific waters

The North Atlantic species and type species of the genus Terebellides, T. stroemii Sars, 1835, was also widely reported in the Indo-Pacific area. For instance, in Indonesia by Caullery (1944), in South Korea by Gallardo (1967), in Hong Kong by Shin (1982), in Singapore by Tan & Chou (1993) and in the Australian coasts by Stephenson, Williams & Lance (1970), Stephenson, Williams & Cook (1974), Knox & Cameron (1971), Hutchings (1977), Amoureux (1984), Hutchings & Murray (1984) and Hutchings et al. (1993) (Fig. 7). In the Southern Pacific Ocean, the presence of T. stroemii was denied by Hutchings & Peart (2000) after examining Norwegian material. Indeed, these authors already reassigned part of these Indo-Pacific reports to other species, while others specimens were not. Among them, the material reported by Caullery (1944) and collected during the Siboga expedition might well correspond to more than one species according to the description and illustrations. The shape of the branchiae in specimen from station 271 (Fig. 147 in Caullery, 1944; redrawn here in Fig. 8B) and station 311 (Fig. 148 in Caullery, 1944; redrawn here in Figs. 8C and 8D) sharply differs in BL5 size; the specimen of station 311 is more similar in branchial shape to T. hutchingsae spec. nov. but differs in the high degree of fusion of dorsal and ventral lobes in Caullery’s material (see Fig. 8D). The specimen reported by Gallardo (1967) cannot be properly identified because the description is quite brief (e.g., “The branchia has the typical shape…”) and only a lateral view of a thoracic uncinus is illustrated and this is not relevant in species discrimination. More recently, Parapar & Hutchings (2015) redescribed T. stroemii based on Norwegian specimens collected by Michael Sars near the type locality. Therefore, the type species is now well-illustrated based on the most recent, taxonomically robust characters. This redescription, together with the previous observations and our present results, certainly allows us to strongly support the absence of T. stroemii from Indo-Pacific waters.

Taxonomic relevance of the ciliated papillae of branchial lamellae

One of the most relevant diagnostic characters of T. hutchingsae spec. nov. is the presence of ciliated papillae in branchial lamellae. This character was long ignored in Terebellides descriptions and was discussed by Parapar, Moreira & O’Reilly (2016) and Parapar et al. (2016). In fact, several recently described species from across the world oceans show this feature (Table 2), namely T. gracilis Malm, 1874 sensu Parapar, Moreira & Helgason (2011), off Iceland; T. jorgeni Hutchings, 2007, from Indonesia; T. gracilis Malm, 1874 sensu Parapar, Mikac & Fiege (2013) and T. mediterranea Parapar, Mikac & Fiege, 2013, from the Adriatic Sea; T. akares Hutchings, Nogueira & Carrerette, 2015, from the Great Barrier Reef (NE Australia); T. persiae Parapar et al., 2016, from the Persian Gulf and T. cf. woolawa Hutchings & Peart, 2000 sensu Parapar et al. (this work) from South Myanmar. This character is probably much more widespread and shows at least two different morphotypes: (1) low papillae as it was found in T. gracilis from Iceland and the Mediterranean, and (2) well developed papillae in the rest of species. The presence of these low ciliated papillae reported in Icelandic and Adriatic specimens of T. gracilis by Parapar, Moreira & Helgason (2011) and Parapar, Mikac & Fiege (2013) was confirmed in the holotype. Therefore, these ciliated papillae appear as a robust taxonomic character and we strongly recommend taking it into account in future descriptions of new species within the genus.

We are indebted to the staff members of Créocéan, who recovered the material during the processing of benthic macrofaunal samples and explicitly accepted to allow publishing the information contained in this study, and to Andrea Feijoo (UDC) for the initial study of the specimens. Special thanks are also due to Ada Castro and Cati Sueiro (SAI, UDC) for assisting with the preparation of the specimens and use of the SEM, and to Xela Cunha (Marine Station of A Graña-Ferrol, University of Santiago de Compostela-USC, Spain) for her assistance with the stereomicroscope images. Authors also wish to thank Kennet Lundin (GNM) for sending T. gracilis holotype, and Lena Gustavsson (NRM) for her help finding T. sieboldi holotype. Thanks are also due to João Nogueira, Marcelo Fukuda and André S. Garraffoni for their useful comments, which greatly improve the manuscript.

Abbreviations used in text, tables and figures

BL Branchial lobes

BT Buccal tentacles

CHG Chaetiger with geniculate chaetae

DL Dorsal lobes

GC Geniculate chaeta

GO Genital opening

LL Lateral lappets

NACH Number of abdominal chaetigers

LCP Low ciliated papillae

NPA Nephridial papilla

NRTU Number of rows of frontal rostral teeth in thoracic uncini

PPP Posterior pointed projection

R Rostrum

SG Segment

TC Thoracic chaetiger

TN Thoracic notopodia

TP Terminal projection

VL Ventral lobes

WDCP Well developed ciliated papillae

Additional Information and Declarations

Competing Interests

Author Contributions

Data Availability

New Species Registration

The authors declare there are no competing interests.

Julio Parapar and Juan Moreira performed the experiments, analyzed the data, contributed reagents/materials/analysis tools, wrote the paper, prepared figures and/or tables, reviewed drafts of the paper.

Daniel Martin conceived and designed the experiments, performed the experiments, analyzed the data, contributed reagents/materials/analysis tools, wrote the paper, prepared figures and/or tables, reviewed drafts of the paper.

The following information was supplied regarding data availability:

The raw data is contained in Table 1.

The following information was supplied regarding the registration of a newly described species:

Publication LSID: urn:lsid:zoobank.org:pub:39745D2F-9163-48B2-9FAB- FBF66D3AEFB5

Terebellides hutchingsae species LSID: urn:lsid:zoobank.org:act:78E96984-41E7-43E6-8E5D-03E9421BE306.

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
