# Peer review of "On the diversity of the SE Indo-Pacific species of Terebellides (Annelida; Trichobranchidae), with the description of a new species"

_PeerJ, doi:10.7717/peerj.2313_

## Round 0.1 · original submission · Minor Revisions

Dear Dani,

Congratulations on a nice manuscript. Please address all the minor issues raised by the three reviewers (most comments are in the three manuscripts they returned). The most important issues to address are the citations and the abbreviations. Please include all references in the literature cited, including taxonomic authorities. Please check the list of abbreviations provided in the Materials and Methods and be sure all are there. Also, please remove all abbreviations from the key and wherever possible throughout the manuscript within reason to improve readability. Also, include abbreviations in the figure legends when they are included in the figures.

Cheers - Karen

·

Basic reporting

All comments below

Experimental design

All comments below

Validity of the findings

All comments below

Additional comments

Thank you for the opportunity for reviewing this nice manuscript. As I said, it is a nice manuscript, with good illustrations and the descriptions are very complete. I think several parts, however, should be improved for a better quality of the final paper. I wrote all my comments in the manuscript file but I could find how too attach it here, please let me know how I can send it to the authors. As explained along that reviews file, I summarise my questions under four main points:
1- I couldn't find figure and table legends, and I suggest the authors include the meaning of the letter legends in all figures, in order to facilitate the understanding of these figures (it is really too bad if one has to go to M&M everytime he wants to understand one photo);
2- Authors use chaetiger number instead of segment numbers to describe several characters. However, a few species of Terebellides have notopodia beginning on segment 4, instead of segment 3. This means that a change on chaetiger number for several structures (for instance, beginning of neuropodia, position of nephridial papillae, and several others) does necessarily not mean a change on segment number. Because of that, I recommend the authors include segment numbers for every structure; it can be between parentheses, as authors do in some parts;
3- Authors should write in full the characters used for the key. Again, it is too bad having to go back and forth between M&M and the key to understand what authors mean. Also, there are some parts I could not understand, some key steps looked meaningless to me and I just gave up checking the steps. The way the key is, I consider it almost impossible to follow and I strongly recommend the authors check it;
4- I don't know if the style of the journal asks to include authorities who described taxa in the References or not, so I leave this to the editor and the authors. But the References section needs a careful review anyway. Several references were not cited in the text, others were cited but not included in the references, and many references are included out of the proper alphabetic order.

·

Basic reporting

No comments

Experimental design

No comments

Validity of the findings

No comments

Additional comments

My compliments for one more important paper showing how thourough, well-done morphological studies are, in many cases, enough to differentiate specimens of species with arguible worldwide distributions.
Comments and suggestions in the ms.

Best wishes,
MVF

·

Basic reporting

I am not a native English speaker, but the article reads is smoothly in most of the time. I pointed out two sentences that need to be rephrased.

Experimental design

no comments

Validity of the findings

Indeed a very interesting piece of work dealing with the common genus Terebellides. The new species Terebellides hutchingsae is described, and presented together with some new information about T. aff. Woodlawa. Comments on a putatively new species desingnated Terebellides sp. is also given.

Additional comments

I have some minor comments and suggestions that should be considered by the authors, and so I recommend this interesting manuscript for publication in PeerJ.

---

## Round 0.2 · Minor Revisions

Thank you for your improvements on the manuscript and fully addressing the majority of the issues raised. Two issues remains to be resolved. The suggestion to compare your new species to all others with very similar morphology regardless of type location should be addressed before it can be accepted. Considering the number of opportunities for transport from different regions it is appropriate to compare to all other similar species, not just those local to your focus. Additionally, description of the staining patterns should be included as well. Validity of that character can only be assessed if data is available for a wide variety of species and range of specimens within each species and at the time of description is the ideal time to be as thorough as possible. You have a large type set and are well positioned to provide useful data for this issue.

---

## Round 0.3 · accepted · Accept

Nice work, thank you for addressing the last details.